# SINGLE-TRAJECTORY DISTRIBUTIONALLY ROBUST REINFORCEMENT LEARNING

## ABSTRACT

To mitigate the limitation that the classical reinforcement learning (RL) framework heavily relies on identical training and test environments, Distributionally Robust RL (DRRL) has been proposed to enhance performance across a range of environments, possibly including unknown test environments. As a price for robustness gain, DRRL involves optimizing over a set of distributions, which is inherently more challenging than optimizing over a fixed distribution in the nonrobust case. Existing DRRL algorithms are either model-based or fail to learn from a single sample trajectory. In this paper, we design the first fully model-free DRRL algorithm, called *distributionally robust Q-learning with single trajectory (DRQ)*. We delicately design a multi-timescale framework to fully utilize each incrementally arriving sample and directly learn the optimal distributionally robust policy without modeling the environment, thus the algorithm can be trained along a single trajectory in a model-free fashion. Despite the algorithm's complexity, we provide asymptotic convergence guarantees by generalizing classical stochastic approximation tools. Comprehensive experimental results demonstrate the superior robustness and sample complexity of our proposed algorithm, compared to non-robust methods and other robust RL algorithms.

## 1 INTRODUCTION

Reinforcement Learning (RL) is a machine learning paradigm for studying sequential decision problems. Despite considerable progress in recent years (Silver et al., 2016; Mnih et al., 2015; Vinyals et al., 2019), RL algorithms often encounter a discrepancy between training and test environments. This discrepancy is widespread since test environments may be too complex to be perfectly represented in training, or the test environments may inherently shift from the training ones, especially in certain application scenarios, such as financial markets and robotic control. Overlooking the mismatch could impede the application of RL algorithms in real-world settings, given the known sensitivity of the optimal policy of the Markov Decision Process (MDP) to the model (Mannor et al., 2004; Iyengar, 2005).

To address this concern, Distributionally Robust RL (DRRL) (Zhou et al., 2021; Yang et al., 2022; Shi & Chi; Panaganti & Kalathil, 2022; Panaganti et al., 2022; Ma et al., 2022; Yang, 2018; Abdullah et al., 2019; Neufeld & Sester, 2022) formulates the decision problem under the assumption that the test environment varies but remains close to the training environment. The objective is to design algorithms optimizing the worst-case expected return over an ambiguity set encompassing all possible test distributions. Evaluating a DRRL policy necessitates deeper insight into the transition dynamics than evaluating a non-robust one, as it entails searching for the worst-case performance across all distributions within the ambiguity set. Therefore, most prior solutions are model-based, require the maintenance of an estimator for the entire transition model and the ambiguity set. Such requirements may render these algorithms less practical in scenarios with large state-action spaces or where adequate modeling of the real environment is unfeasible.

Prompted by this issue, we study a fully model-free DRRL algorithm in this paper, which learns the optimal DR policy without explicit environmental modeling. The algorithm's distinctive feature is its capacity to learn from a single sample trajectory, representing the least demanding requirement for data collection. This feature results from our innovative algorithmic framework, comprising incrementally updated estimators and a delicate approximation scheme. While most model-free

non-robust RL algorithms support training in this setting—contributing to their widespread use—no existing work can effectively address the DRRL problem in this way. The challenge arises from the fact that approximating a DR policy by learning from a single trajectory suffers from restricted control over state-action pairs and limited samples, i.e., only one sample at a time. As we will demonstrate, a simple plug-in estimator using one sample, which is unbiased in the non-robust $Q$-learning algorithm, fails to approximate any robust value accurately.

The complexity of this task is further affirmed by the sole attempt to develop a model-free DRRL algorithm in (Liu et al., 2022). It relies on a restricted simulator assumption, enabling the algorithm to access an arbitrary number of samples from any state-action pair, thereby amassing sufficient system dynamics information before addressing the DRRL problem. Relaxing the dependence on a simulator and developing a fully model-free algorithm capable of learning from a single trajectory necessitates a delicate one-sample estimator for the DR value, carefully integrated into an algorithmic framework to eradicate bias from insufficient samples and ensure convergence to the optimal policy. Moreover, current solutions heavily depend on the specific divergence chosen to construct the ambiguity set and fail to bridge different divergences, underscoring the practical importance of divergence selection.

Thus a nature question arises: *Is it possible to develop a model-free DRRL framework that can learn the optimal DR policy across different divergences using only a single sample trajectory for learning?*

## 1.1 OUR CONTRIBUTIONS

In this paper, we provide a positive solution to the aforementioned question by making the following contributions:

1. We introduce a pioneering approach to construct the ambiguity set using the Cressie-Read family of $f$-divergence. By leveraging the strong duality form of the corresponding distributionally robust reinforcement learning (DRRL) problem, we reformulate it, allowing for the learning of the optimal DR policies using misspecified MDP samples. This formulation effortlessly covers widely used divergences such as the Kullback-Leibler (KL) and $\chi^2$ divergence.

2. To address the additional nonlinearity that arises from the DR Bellman equation, which is absent in its non-robust counterpart, we develop a novel multi-timescale stochastic approximation scheme. This scheme carefully exploits the structure of the DR Bellman operator. The update of the $Q$ table occurs in the slowest loop, while the other two loops are delicately designed to mitigate the bias introduced by the plug-in estimator due to the nonlinearity.

3. We instantiate our framework into a DR variant of the $Q$-learning algorithm, called *distributionally robust Q-learning with single trajectory (DRQ)*. This algorithm solves discount Markov Decision Processes (MDPs) in a fully online and incremental manner. We prove the asymptotic convergence of our proposed algorithm by extending the classical two-timescale stochastic approximation framework, which may be of independent interest.

4. We conduct extensive experiments to showcase the robustness and sample efficiency of the policy learned by our proposed DR $Q$-learning algorithm. We also create a deep learning version of our algorithm and compare its performance to representative online and offline (robust) reinforcement learning benchmarks on classical control tasks.

## 1.2 RELATED WORK

**Robust MDPs and RL:** The framework of robust MDPs has been studied in several works such as Nilim & El Ghaoui (2005); Iyengar (2005); Wiesemann et al. (2013); Lim et al. (2013); Ho et al. (2021); Goyal & Grand-Clement (2022). These works discuss the computational issues using dynamic programming with different choices of MDP formulation, as well as the choice of ambiguity set, when the transition model is known. Robust Reinforcement Learning (RL) (Roy et al., 2017; Badrinath & Kalathil, 2021; Wang & Zou, 2021) relaxes the requirement of accessing to the transition model by simultaneously approximating to the ambiguity set as well as the optimal robust policy, using only the samples from the misspecified MDP.

**Online Robust RL:** Existing online robust RL algorithms including Wang & Zou (2021); Badrinath & Kalathil (2021); Roy et al. (2017), highly relies on the choice of the $R$-contamination model and could suffer over-conservatism. This ambiguity set maintains linearity in their corresponding Bellman

operator and thus inherits most of the desirable benefits from its non-robust counterpart. Instead, common distributionally robust ambiguity sets, such as KL or $\chi^2$ divergence ball, suffer from extra nonlinearity when trying to learn along a single-trajectory data, which serves as the foundamental challenge in this paper.

**Distributionally Robust RL:** To tackle the over-conservatism aroused by probability-agnostic $R$-contamination ambiguity set in the aforementioned robust RL, DRRL is proposed by constructing the ambiguity set with probability-aware distance (Zhou et al., 2021; Yang et al., 2022; Shi & Chi; Panaganti & Kalathil, 2022; Panaganti et al., 2022; Ma et al., 2022), including KL and $\chi^2$ divergence. As far as we know, most of the existing DRRL algorithms fall into the model-based fashion, which first estimate the whole transition model and then construct the ambiguity set around the model. The DR value and the corresponding policy are then computed based upon them. Their main focus is to understand the sample complexity of the DRRL problem in the offline RL regime, leaving the more prevalent single-trajectory setting largely unexplored.

## 2 PRELIMINARY

### 2.1 DISCOUNTED MDPs

Consider an infinite-horizon MDP $(\mathcal{S}, \mathcal{A}, \gamma, \mu, P, r)$ where $\mathcal{S}$ and $\mathcal{A}$ are finite state and action spaces with cardinality $S$ and $A$. $P : \mathcal{S} \times \mathcal{A} \to \Delta_S$ is the state transition probability measure. Here $\Delta_S$ is the set of probability measures over $\mathcal{S}$. $r$ is the reward function and $\gamma$ is the discount factor. We assume that $r : \mathcal{S} \times \mathcal{A} \to [0, 1]$ is deterministic and bounded in $[0, 1]$. A stationary policy $\pi : \mathcal{S} \to \Delta_A$ maps, for each state $s$ to a probability distribution over the action set $\mathcal{A}$ and induce a random trajectory $s_1, a_1, r_1, s_2, \cdots$, with $s_1 \sim \mu$, $a_n = \pi(s_n)$ and $s_{n+1} \sim P(\cdot|s_n, a_n) := P_{s_n, a_n}$ for $n \in \mathbb{N}^+$. To derive the policy corresponding to the value function, we define the optimal state-action function $Q^\star : \mathcal{S} \times \mathcal{A} \to \mathbb{R}$ as the expected cumulative discounted rewards under the optimal policy, $Q^\star(s, a) := \sup_{\pi \in \Pi} \mathbb{E}_{\pi, P}[\sum_{n=1}^{\infty} \gamma^{n-1} r(s_n, a_n)|s_1 = s, a_1 = a]$. The optimal state-action function $Q^*$ is also the fixed point of the Bellman optimality equation,

$$Q^\star(s, a) = r(s, a) + \gamma \mathbb{E}_{s' \sim P}[\max_{a' \in \mathcal{A}} Q^\star(s', a')]. \tag{1}$$

### 2.2 $Q$-LEARNING

Our model-free algorithmic design relies on the $Q$-learning template, originally designed to solve the non-robust Bellman optimality equation (Equation 1). $Q$-learning is a model-free reinforcement learning algorithm that uses a single sample trajectory to update the estimator for the $Q$ function incrementally. Suppose at time $n$, we draw a sample $(s_n, a_n, r_n, s'_n)$ from the environment. Then, the algorithm updates the estimated $Q$-function following:

$$Q_{n+1}(s_n, a_n) = (1 - \alpha_n) Q_n(s_n, a_n) + \alpha_n(r_n + \gamma \max_{a' \in \mathcal{A}} Q_n(s'_n, a')),$$

Here, $\alpha_n > 0$ is a learning rate. The algorithm updates the estimated $Q$ function by constructing a unbiased estimator for the true $Q$ value, i.e., $r_n + \gamma \max_{a' \in \mathcal{A}} Q_n(s'_n, a')$ using one sample.

### 2.3 DISTRIBUTIONALLY ROBUST MDPs

DRRL learns an optimal policy that is robust to unknown environmental changes, where the transition model $P$ and reward function $r$ may differ in the test environment. To focus on the perturbation of the transition model, we assume no pertubation to the reward function. Our approach adopts the notion of distributional robustness, where the true transition model $P$ is unknown but lies within an ambiguity set $\mathcal{P}$ that contains all transition models that are close to the training environment under some probability distance $D$. To ensure computational feasibility, we construct the ambiguity set $\mathcal{P}$ in the $(s, a)$-rectangular manner, where for each $(s, a) \in \mathcal{S} \times \mathcal{A}$, we define the ambiguity set $\mathcal{P}_{s,a}$ as,

$$\mathcal{P}_{s,a} := \{P'_{s,a} : \Delta_S | D(P'_{s,a} \| P_{s,a}) \le \rho\}. \tag{2}$$

We then build the ambiguity set for the whole transition model as the Cartesian product of every $(s, a)$-ambiguity set, i.e., $\mathcal{P} = \prod_{(s,a) \in \mathcal{S} \times \mathcal{A}} \mathcal{P}_{s,a}$. Given $\mathcal{P}$, we define the optimal DR state-action

function $Q^\star$ as the value function of the best policy to maximize the worst-case return over the ambiguity set,

$$Q^{\mathrm{rob},\star}(s,a) := \sup_{\pi \in \Pi} \inf_{P \in \mathcal{P}} \mathbb{E}_{\pi,P}[\sum_{n=1}^{\infty} \gamma^{n-1} r(s_n, a_n)|s_1 = s, a_1 = a].$$

Under the $(s,a)$-rectangular assumption, the Bellman optimality equation has been established by Iyengar (2005); Xu & Mannor (2010),

$$Q^{\mathrm{rob},\star}(s,a) = \mathcal{T}_k(Q^{\mathrm{rob},\star})(s,a) := r(s,a) + \gamma \inf_{P \in \mathcal{P}} \mathbb{E}_{s' \sim P}[\max_{a' \in \mathcal{A}} Q^{\mathrm{rob},\star}(s',a')]. \quad (3)$$

For notation simplicity, we would ignore the superscript rob.

## 3 DISTRIBUTONALLY ROBUST $Q$-LEARNING WITH SINGLE TRAJECTORY

This section presents a general model-free framework for DRRL. We begin by instantiating the distance $D$ as Cressie-Read family of $f$-divergence (Cressie & Read, 1984), which is designed to recover previous common choices such as the $\chi^2$ and KL divergence. We then discuss the challenges and previous solutions in solving the corresponding DRRL problem, as described in Section 3.2. Finally, we present the design idea of our three-timescale framework and establish the corresponding convergence guarantee.

### 3.1 DIVERGENCE FAMILIES

Previous work on DRRL has mainly focused on one or several divergences, such as KL, $\chi^2$, and total variation (TV) divergences. In contrast, we provide a unified framework that applies to a family of divergences known as the Cressie-Read family of $f$-divergences. This family is parameterized by $k \in (-\infty, \infty)/\{0, 1\}$, and for any chosen $k$, the Cressie-Read family of $f$-divergences is defined as

$$D_{f_k}(Q\|P) = \int f_k(\frac{dP}{dQ})dQ, \ \text{with } f_k(t) := \frac{t^k - kt + k - 1}{k(k-1)}.$$

Based on this family, we instantiate our ambiguity set in Equation 2 as $\mathcal{P}_{s,a} = \{P'_{s,a} : \Delta_S | D_{f_k}(P'_{s,a}\|P_{s,a}) \le \rho\}$ for some radius $\rho > 0$. The Cressie-Read family of $f$-divergence includes $\chi^2$-divergence ($k = 2$) and KL divergence ($k \to 1$).

One key challenge in developing DRRL algorithms using the formulation in Equation 3 is that the expectation is taken over the ambiguity set $\mathcal{P}$, which is computationally intensive even with the access to the center model $P$. Since we only have access to samples generated from the possibly misspecific model $P$, estimating the expectation with respect to other models $P' \in \mathcal{P}$ is even more challenging. While importance sampling-based techniques can achieve this, the cost of high variance is still undesirable. To solve this issue, we rely on the dual reformulation of Equation 3:

**Lemma 3.1** (Duchi & Namkoong (2021)). *For any random variable $X \sim P$, define $\sigma_k(X, \eta) = -c_k(\rho)\mathbb{E}_P[(\eta - X)_+^{k_*}]^{\frac{1}{k_*}} + \eta$ with $k_* = \frac{k}{k-1}$ and $c_k(\rho) = (1 + k(k-1)\rho)^{\frac{1}{k}}$. Then*

$$\inf_{Q \ll P}\{\mathbb{E}_Q[X] : D_{f_k}(Q\|P) \le \rho\} = \sup_{\eta \in \mathbb{R}} \sigma_k(X, \eta), \quad (4)$$

Here $(x)_+ = \max\{x, 0\}$. Equation 4 shows that protecting against the distribution shift is equivalent to optimizing the tail-performance of a model, as only the value below the dual variable $\eta$ are taken into account. Another key insight from the reformulation is that as the growth of $f_k(t)$ for large $t$ becomes steeper for larger $k$, the $f$-divergence ball shrinks and the risk measure becomes less conservative. This bridges the gap between difference divergences, whereas previous literature, including Yang et al. (2022) and Zhou et al. (2021), treats different divergences as separate. By applying the dual reformulation, we can rewrite the Cressie-Read Bellman operator in Equation 3 as

$$\mathcal{T}_k(Q)(s,a) = r(s,a) + \gamma \sup_{\eta \in \mathbb{R}} \sigma_k(\max_{a' \in \mathcal{A}} Q(\cdot, a'), \eta). \quad (5)$$

### 3.2 Bias in Plug-in Estimator in Single Trajectory Setting

In this subsection, we aim to solve Equation 5 using single-trajectory data, which has not been addressed by previous DRRL literature. As we can only observe one newly arrival sample each time, to design a online model-free DRRL algorithm, we need to approximate the expectation in Equation 5 using that single sample properly. As mentioned in Section 2.2, the design of the $Q$-learning algorithm relies on an one-sample unbiased estimator of the true Bellman operator. However, this convenience vanishes in the DR Bellman operator. To illustrate this, consider plugging only one sample into the Cressie-Read Bellman operator Equation 5:

$$r(s, a) + \gamma \sup_{\eta \in \mathbb{R}} \{\eta - c_k(\rho)(\eta - \max_{a'} Q(s', a'))_+\} = r(s, a) + \gamma \max_{a'} Q(s', a').$$

This reduces to the non-robust Bellman operator and is obviously not an unbiased estimator for $\mathcal{T}_k(Q)$. This example reveals the inherently more challenging nature of the online DRRL problem. Whereas non-robust RL only needs to improve the expectation of the cumulative return, improving the worst-case return requires more information about the system dynamics, which seems hopeless to be obtained from only one sample and sharply contrasts with our target.

Even with the help of batch samples, deriving an appropriate estimator for the DR Bellman operator is still nontrivial. Consider a standard approach to construct estimators, sample average approximation (SAA): given a batch of sample size $n$ starting from a fix state-action pair $(s, a)$, i.e., $D_n = \{(s_i, a_i, s'_i, r_i), i \in [n], (s_i, a_i) = (s, a)\}$, the SAA empirical Bellman operator is defined as:

$$\widehat{\mathcal{T}}_k(Q)(s, a, D_n) = r(s, a) + \gamma \sup_{\eta \in \mathbb{R}} \widehat{\sigma}_k(\max_{a' \in \mathcal{A}} Q(\cdot, a'), \eta, D_n)$$

$$= r(s, a) + \sup_{\eta \in \mathbb{R}} \{-c_k(\rho)[\sum_{i \in [n]} (\eta - \max_{a' \in \mathcal{A}} Q(s'_i, a'))_+^{k_*}/n]^{\frac{1}{k_*}} + \eta\}$$

Here, $\widehat{\sigma}_k$ is the empirical Cressie-Read functional. As pointed out by Liu et al. (2022), the SAA estimator is biased, prompting the introduction of the multilevel Monte-Carlo method (Blanchet & Glynn, 2015). Specifically, it first obtains $N \in \mathbb{N}^+$ samples from the distribution $\mathbb{P}(N = n) = p_n = \epsilon(1 - \epsilon)^n$, and then uses the simulator to draw $2^{N+1}$ samples $D_{2^{N+1}}$. The samples are further decomposed into two parts: $D_{:2^N}$ consists of the first $2^N$ samples, while $D_{2^N+1:}$ contains the remaining samples. Finally, the DR term in Equation 5 is approximated by solving three optimization problems:

$$\widehat{\mathcal{T}}_k(Q)(s, a, D_n) = r_1 + \max_{a' \in \mathcal{A}} Q(s'_1, a') + \frac{\Delta_{N, \delta}^q(Q)}{p_N},$$

where $\Delta_{N, \delta}^q(Q) \coloneqq \sup_{\eta \geq 0} \widehat{\sigma}_k(\max_{a' \in \mathcal{A}} Q(\cdot, a'), \eta, D_{2^{N+1}})$
$-\frac{1}{2} \sup_{\eta \geq 0} \widehat{\sigma}_k(\max_{a' \in \mathcal{A}} Q(\cdot, a'), \eta, D_{:2^N}) - \frac{1}{2} \sup_{\eta \geq 0} \widehat{\sigma}_k(\max_{a' \in \mathcal{A}} Q(\cdot, a'), \eta, D_{2^N+1:}).$

However, this multilevel Monte-Carlo solution requires a large batch of samples for the same state-action pair before the next update, resulting in unbounded memory costs/computational time that are not practical. Furthermore, it is prohibited in the single-trajectory setting, where each step only one sample can be observed. Our experimental results show that simply approximating the Bellman operator with simulation data, without exploiting its structure, suffers from low data efficiency.

### 3.3 Three-timescale Framework

The $Q$-learning is solving the nonrobust Bellman operator's fixed point in a stochastic approximation manner. A salient feature in the DR Bellman operator, compared with its nonrobust counterpart, is a bi-level optimization nature, i.e., jointly solving the dual parameter $\eta$ and the fixed point $Q$ of the Bellman optimality equation. We revisit the stochastic approximation view of the $Q$-learning and develop a three-timescale framework, by a faster running estimate of the optimal dual parameter, and a slower update of the $Q$ table. To solve Equation 5 using a stochastic approximation template, we iteratively update the variables $\eta$ and $Q$ table as follows: for the $n$-th iteration after observing a new transition sample $(s_n, a_n, s'_n, r_n)$ and some learning rates $\zeta_1, \zeta_2 > 0$,

$$\eta_{n+1} = \eta_n - \zeta_1 * \text{Gradient of } \eta_n,$$

$$Q_{n+1} = r_n + \zeta_2 \gamma \sigma_k(\max_{a' \in \mathcal{A}} Q_n(\cdot, a'), \eta_n).$$

As the update of $\eta$ and $Q$ relies on each other, we keep the learning speeds of $\eta$ and $Q$, i.e., $\zeta_1$ and $\zeta_2$, different to stabilize the training process. Additionally, due to the $(s, a)$-rectangular assumption, $\eta$ is independent across different $(s, a)$-pairs, while the $Q$ table depends on each other. The independent structure for $\eta$ allows it to be estimated more easily; so we approximate it in a faster loop, while for $Q$ we update it in a slower loop. To instantiate this template further, we compute the gradient of $\sigma_k(\max_{a' \in \mathcal{A}} Q(\cdot, a'), \eta)$ in Equation 5 with respect to $\eta$.

---

**Algorithm 1** Distributionally Robust $Q$-learning with Cressie-Read family of $f$-divergences

---

1: **Input:** Exploration rate $\epsilon$, Learning rates $\{\zeta_i(n)\}_{i \in [3]}$, Cressie-Read family parameter $k$, Ambiguity set radius $\rho$.
2: **Init:** Initialize $Q$, $Z$ and $\eta$ with zero.
3: **for** $n = 1, 2, \cdots$ **do**
4:     Observe the state $s_n$, execute the action $a_n = \arg\max_{a \in \mathcal{A}} Q(s_n, a)$ using $\epsilon$-greedy policy
5:     Observe the reward $r_n$ and next state $s'_n$
6:     Update $Z_1(s_n, a_n) \leftarrow (1 - \zeta_1(n))Z_1(s_n, a_n) + \zeta_1(n)(\eta(s_n, a_n) - \max_a Q(s'_n, a))_+^{k_*}$, and
       $Z_2(s_n, a_n) \leftarrow (1 - \zeta_1(n))Z_2(s_n, a_n) + \zeta_1(n)(\eta(s_n, a_n) - \max_a Q(s'_n, a))_+^{k_*-1}$.
7:     Update $\eta(s_n, a_n) \leftarrow \eta(s_n, a_n) + \zeta_2(n)(-c_k(\rho)Z_1^{\frac{1}{k_*}-1}(s_n, a_n) \cdot Z_2(s_n, a_n) + 1)$.
8:     Update $Q(s_n, a_n) \leftarrow (1 - \zeta_3(n))Q(s_n, a_n) + \zeta_3(n)(r_n - \gamma(c_k(\rho)Z_1^{\frac{1}{k_*}}(s_n, a_n) - \eta(s_n, a_n)))$.
9: **end for**

---

**Lemma 3.2** (Gradient of the $\sigma_k$ dual function).

$$\frac{d\sigma_k}{d\eta}(\max_{a' \in \mathcal{A}} Q(\cdot, a'), \eta) = -c_k(\rho)Z_1^{\frac{1}{k_*}-1} \cdot Z_2 + 1, \tag{6}$$

*where*

$$Z_1 = \mathbb{E}_P[(\eta - \max_{a' \in \mathcal{A}} Q(\cdot, a'))_+^{k_*}], \quad Z_2 = \mathbb{E}_P[(\eta - \max_{a' \in \mathcal{A}} Q(\cdot, a'))_+^{k_*-1}]. \tag{7}$$

Due to the nonlinearity in Equation 6, the plug-in gradient estimator is in fact biased. The bias arises as for a random variable $X$, $\mathbb{E}[f(X)] \neq f(\mathbb{E}[X])$ for $f(x) = x^{\frac{1}{k_*}-1}$ in $Z_1^{\frac{1}{k_*}-1}$. To address this issue, we introduce another even faster timescale to estimate $Z_1$ and $Z_2$,

$$Z_1(s_n, a_n) \leftarrow (1 - \zeta_1(n))Z_1(s_n, a_n) + \zeta_1(n)(\eta(s_n, a_n) - \max_{a'} Q(s'_n, a'))_+^{k_*}, \tag{8}$$

$$Z_2(s_n, a_n) \leftarrow (1 - \zeta_1(n))Z_2(s_n, a_n) + \zeta_1(n)(\eta(s_n, a_n) - \max_{a'} Q(s'_n, a'))_+^{k_*-1}. \tag{9}$$

In the medium timescale, we approximate $\eta^\star(s, a) := \arg\max_{\eta \in \mathcal{R}} \sigma_k(\max_{a' \in \mathcal{A}} Q(s, a'), \eta)$ via

$$\eta(s_n, a_n) \leftarrow \eta(s_n, a_n) + \zeta_2(n)(-c_k(\rho)Z_1^{\frac{1}{k_*}-1}(s_n, a_n) \cdot Z_2(s_n, a_n) + 1). \tag{10}$$

Finally, we update the DR $Q$ function in the slowest timescale using Equation 11,

$$Q(s_n, a_n) \leftarrow (1 - \zeta_3(n))Q(s_n, a_n) + \zeta_3(n)\widehat{\mathcal{T}}_{n,k}(Q)(s_n, a_n), \tag{11}$$

where $\widehat{\mathcal{T}}_{n,k}(Q)(s, a)$ is the empirical version of Equation 5 in the $n$-th iteration:

$$\widehat{\mathcal{T}}_{n,k}(Q)(s_n, a_n) = r_n - \gamma(c_k(\rho)Z_1^{\frac{1}{k_*}}(s_n, a_n) - \eta(s_n, a_n)).$$

Here $\zeta_1(n), \zeta_2(n)$ and $\zeta_3(n)$ are learning rates for three timescales at time $n$, which will be specified later. We summarize the ingredients into our DR $Q$-learning (DRQ) algorithm (Algorithm 1), and prove the almost surely (a.s.) convergence of the algorithm as Theorem 3.3. The proof is deferred in Appendix D.

**Theorem 3.3.** *The estimators at the $n$-th step in Algorithm 1, $(Z_{n,1}, Z_{n,2}, \eta_n, Q_n)$, converge to $(Z_1^\star, Z_2^\star, \eta^\star, Q^\star)$ a.s. as $n \to \infty$, where $\eta^\star$ and $Q^\star$ are the fixed-point of the equation $Q = \mathcal{T}_k(Q)$, and $Z_1^\star$ and $Z_2^\star$ are the corresponding quantity under $\eta^\star$ and $Q^\star$.*

The proof establishes that, by appropriately selecting stepsizes to prioritize frequent updates of $Z_{n,1}$ and $Z_{n,2}$, followed by $\eta_n$, and with $Q_n$ updated at the slowest rate, the solution path of $(Z_{n,1}, Z_{n,2}, \eta_n, Q_n)$ closely tracks a system of three-dimensional ordinary differential equations (ODEs) considering martingale noise. Our approach is to generalize the classic machinery of two-timescale stochastic approximation (Borkar, 2009) to a three-timescale framework, and use it to analyze our proposed algorithm. See Appendix C for the detailed proof.

## 4 EXPERIMENTS

We demonstrate the robustness and sample complexity of our DRQ algorithm in the Cliffwalking environment (Delétang et al., 2021) and American put option environment (deferred in Appendix B). These environments provide a focused perspective on the policy and enable a clear understanding of the key parameters effects. We develop a deep learning version of DRQ and compare it with practical online and offline (robust) RL algorithms in classical control tasks, LunarLander and CartPole.

### 4.1 CONVERGENCE AND SAMPLE COMPLEXITY

Before we begin, let us outline the key findings and messages conveyed in this subsection: **(1) Our ambiguity set design provides substantial robustness**, as demonstrated through comparisons with non-robust $Q$-learning and $R$-contamination ambiguity sets (Wang & Zou, 2021). **(2) Our DRQ algorithm exhibits desirable sample complexity**, significantly outperforming the multi-level Monte Carlo based DRQ algorithm proposed by Liu et al. (2022) and comparable to the sample complexity of the model-based DRRL algorithm by Panaganti & Kalathil (2022).



| (a) Environment | (b) Nonrobust | (c) $\rho = 0.5$ | (d) $\rho = 1.0$ | (e) $\rho = 1.5$ |

Figure 1: The Cliffwalking environment and the learned policies for different $\rho$'s.

**Experiment Setup:** The Cliffwalking task is commonly used in risk-sensitive RL research (Delétang et al., 2021). Compared to the Frozen Lake environment used by Panaganti & Kalathil (2022), Cliffwalking offers a more intuitive visualization of robust policies (see Figure 1). The task involves a robot navigating from an initial state of $(2, 0)$ to a goal state of $(2, 3)$. At each step, the robot is affected by wind, which causes it to move in a random direction with probability $p$. Reaching the goal state earns a reward of $+5$, while encountering a wave in the water region $\{(3, j) \mid 0 \leq j \leq 3\}$ results in a penalty of $-1$. We train the agent in the nominal environment with $p = 0.5$ for 3 million steps per run, using an $\epsilon$-greedy exploration strategy with $\epsilon = 0.1$. We evaluate its performance in perturbed environments, varying the choices of $k$ and $\rho$ to demonstrate different levels of robustness. We set the stepsize parameters according to Assumption C.1: $\zeta_1(t) = 1/(1 + (1 - \gamma)t^{0.6})$, $\zeta_2(t) = 1/(1 + 0.1(1 - \gamma)t^{0.8})$, and $\zeta_3(t) = 1/(1 + 0.05(1 - \gamma) * t)$, where the discount factor is $\gamma = 0.9$.

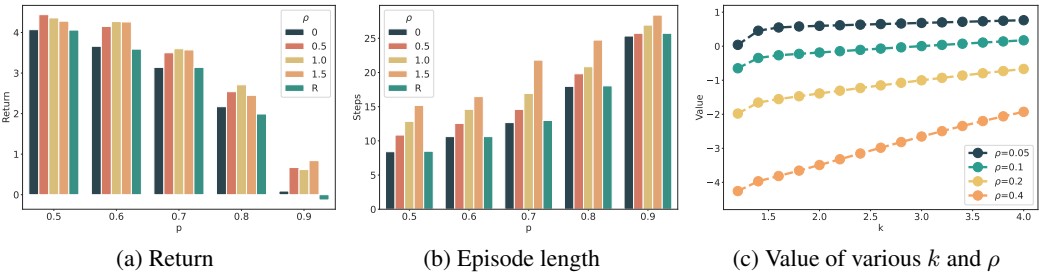

| (a) Return | (b) Episode length | (c) Value of various $k$ and $\rho$ |

Figure 2: Averaged return and steps with 100 random seeds in the perturbed environments. $\rho = 0$ corresponds to the non-robust $Q$-learning. $R$ denotes the $R$-contamination ambiguity set.

**Robustness:** To evaluate the robustness of the learned policies, we compare their cumulative returns in perturbed environments with $p \in \{0.5, 0.6, 0.7, 0.8, 0.9\}$ over 100 episodes per setting. We visulize the decision at each status in Figure 1 with different robustness level $\rho$. In particular, the more robust policy tends to avoid falling into the water, thus arrives to the goal state with a longer path by keeping going up before going right. Figure 2a shows the return distribution for each policy. Figure 2b displays the time taken for the policies to reach the goal, and the more robust policy tends to spend more time, which quantitatively supports our observations in Figure 1. Interestingly, we

find that the robust policies outperform the nonrobust one even in the nominal environment. For the different $\rho$'s, $\rho = 1.0$ is the best within a relatively wide range ($p \in \{0.6, 0.7, 0.8\}$), while $\rho = 1.5$ is preferred in the environment of extreme perturbation ($p = 0.9$). This suggests that DRRL provides a elegant trade-off for different robustness preferences.

We also compare our model-free DRRL algorithm with the robust RL algorithm presented in Wang & Zou (2021), which also supports training using a single trajectory. The algorithm in Wang & Zou (2021) uses an $R$-contamination ambiguity set. We select the best value of $R$ from $0.1$ to $0.9$ and other detailed descriptions in Appendix B. In most cases, the $R$-contamination based algorithm performs very similarly to the non-robust benchmark, and even performs worse in some cases (i.e., $p = 0.8$ and $0.9$), due to its excessive conservatism. As we mentioned in Section 3.1, larger $k$ would render the the risk measure less conservative and thus less sensitive to the change in the ball radius $\rho$, which is empirically confirmed by Figure 2c.

**Sample Complexity:** The training curves in Figure 3 depict the estimated value $\max_a \widehat{Q}(s_0, a)$ (solid line) and the optimal robust value $V^*(s_0)$ (dashed line) for the initial state $s_0$. The results indicate that the estimated value converges quickly to the optimal value, regardless of the values of $k$ and $\rho$. Importantly, our DRQ algorithm achieves a similar convergence rate to the non-robust baseline (represented by the black line). We further compare our algorithm with two robust baselines: the DRQ algorithm with a weak simulator proposed by Liu et al. (2022) (referred to as *Liu's*), and the model-based algorithm introduced by Panaganti & Kalathil (2022) (referred to as *Model*) in Figure 4. To ensure a fair comparison, we set the same learning rate, $\zeta_3(t)$, for our DRQ algorithm and the $Q$-table update loop of the Liu's algorithm, as per their recommended choices. Our algorithm converges to the true DR value at a similar rate as the model-based algorithm, while the Liu's algorithm exhibits substantial deviation from the true value and converges relatively slowly. Our algorithm's superior sample efficiency is attributed to the utilization of first-order information to approximate optimal dual variables, whereas Liu's relies on a large amount of simulation data for an unbiased estimator.

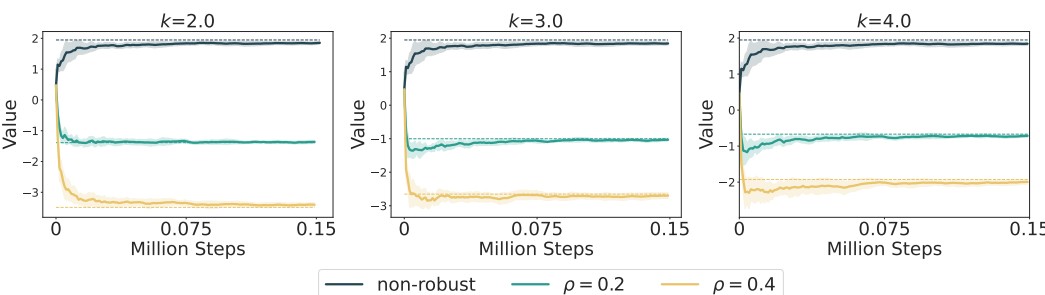

Figure 3: The training curves in the Cliffwalking environment. Each curve is averaged over 100 random seeds and shaded by their standard deviations. The dashed line is the optimal robust value with corresponding $k$ and $\rho$.

## 4.2 PRACTICAL IMPLEMENTATION

We validate the practicality of our DRQ framework by implementing a practical version, called the Deep Distributionally Robust $Q$-learning (*DDRQ*) algorithm, based on the DQN algorithm (Mnih et al., 2015). We apply this algorithm to two classical control tasks from the OpenAI Gym (Brockman et al., 2016): CartPole and LunarLander. Additional experimental details can be found in Appendix B.3.

To assess the effectiveness of our DDRQ algorithm, we compare it against the RFQI algorithm (Panaganti et al., 2022), the soft-robust RL algorithm (Derman et al., 2018), and the non-robust DQN and FQI algorithms. This comparison encompasses representative practical (robust) reinforcement learning algorithms for both online and offline datasets. To evaluate the robustness of the learned policies, we introduce action and physical environment perturbations. For action perturbation, we simulate the perturbations by varying the probability $\epsilon$ of randomly selecting an action for both CartPole and LunarLander tasks. We test with $\epsilon \in \{0, 0.1, 0.2, \cdots, 1.0\}$ for CartPole and $\epsilon \in \{0, 0.1, 0.2, \cdots, 0.6\}$ for LunarLander. Regarding physical environment perturbation in LunarLander, we decrease the power of all the main engine and side engines by the same proportions, ranging from 0

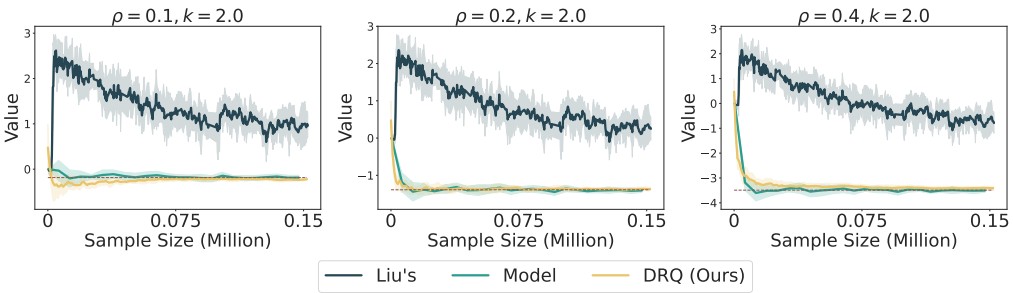

Figure 4: Sample complexity comparisons in Cliffwalking environment with Liu's and Model-based algorithms. Each curve is averaged over 100 random seeds and shaded by their standard deviations.

to 0.6. For CartPole, we reduce the "force mag" parameter from 0.2 to 0.8. We set the same ambiguity set radius for both our DDRQ and RFQI algorithm for fair comparisons. Figure 5 illustrates how our DDRQ algorithm successfully learns robust policies across all tested tasks, achieving comparable performance to other robust counterparts such as RFQI and SR-DQN. Conversely, the non-robust DQN and FQI algorithms fail to learn robust policies and deteriorate significantly even under slight perturbations. It is worth noting that RFQI does not perform well in the LunarLander environment, despite using the official code provided by the authors. This outcome could be attributed to the restriction to their TV distance in constructing the ambiguity set, while our Creass-Read ambiguity set can be flexibly chosen to well adopted to the environment nature. Additionally, the soft-robust RL algorithm requires generating data based on multiple models within the ambiguity set. This process can be excessively time-consuming, particularly in large-scale applications.

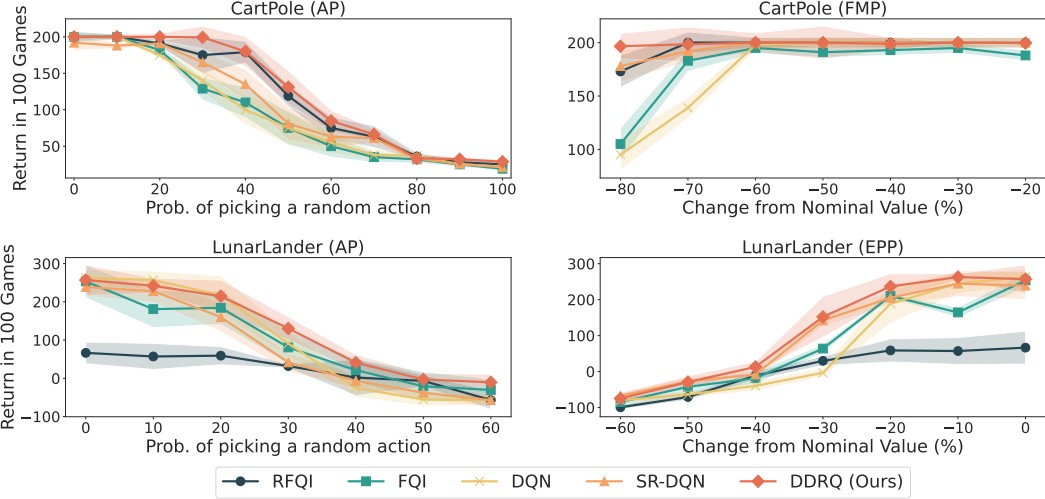

Figure 5: The return in the CartPole and LunarLander environment. Each curve is averaged over 100 random seeds and shaded by their standard deviations. AP: Action Perturbation; FMP: Force Mag Perturbation; EPP: Engines Power Perturbation.

## 5 CONCLUSION

In this paper, we introduce our DRQ algorithm, the first fully model-free DRRL algorithm trained on a single trajectory. By leveraging the stochastic approximation framework, we effectively tackle the joint optimization problem involving the state-action function and the DR dual variable. Through an extension of the classic two-timescale stochastic approximation framework, we establish the asymptotic convergence of our algorithm to the optimal DR policy. Our extensive experimentation showcases the convergence, sample efficiency, and robustness improvements achieved by our approach, surpassing non-robust methods and other robust RL algorithms. Additionally, the deep learning version of our DRQ algorithm validates the practicality of our algorithmic framework.

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
