# OpenReview forum: "Single-Trajectory Distributionally Robust Reinforcement Learning"
_ICLR.cc/2024/Conference — Submitted to ICLR 2024_

### Official Review · Reviewer_7vQe · 2023-10-17

**Soundness:** 2 fair
**Presentation:** 2 fair
**Contribution:** 2 fair
**Rating:** 3
**Confidence:** 4

**Summary:**

This work develops a three time-scale algorithm, aiming to solve the robust RL problem with a single trajectory with an online fashion. The work is important considering the current works in the area.

**Strengths:**

1. The idea is interesting. To solve the unbiased estimation issue, another time scale is introduced. I believe this idea is interesting and novel.
2. The experiment results are promising.

**Weaknesses:**

1. The presentation of the results is infusing and can be improved. E.g., there are too many assumptions made to imply the results. If this work is focusing on the three time-scale stochastic approximation framework itself, it is OK to make these assumptions; But this works is for a concrete problem, i.e., DR-RL, I believe it should not be reasonable to make all these assumptions.
2. As mentioned, with so many assumptions made, I doubt the soundness of the convergence results. E.g., how can I know that under the DR-RL problem, the associated ODE has a unique global asymptotically stable equilibrium?

**Questions:**

1. The work use gradient descent to solve the support function. Why does this DRO problem can be solved by GD? Why is this problem smooth? Why does the gradient exist? I didn't see this gradient descent approach much used in previous DRO works.
2. How do you justify the assumptions you made?

---

> ### Author Response · Authors · 2023-11-20
>
> We appreciate the reviewer's comments and efforts sincerely. We believe the following point-to-point response can address all the concerns and misunderstandings.
>
> ---
>
>  **Q.** Confusing presentation and too many assumptions. How to justify them.
>
> **A.**
> We apologize for the confusing presentation of the results, particularly the organization of the assumptions.
>
> For the main theoretical result (Theorem 3.3), which focuses on the asymptotic convergence of the DRRL algorithm, **we only require ONE assumption about the learning rate (Assumption C.10)**. This assumption can be easily satisfied by some common choices, such as $\zeta_1(n) = \frac{1}{1+n^{0.6}}$, $\zeta_2(n) = \frac{1}{1+n^{0.8}}$, and $\zeta_3(n) = \frac{1}{1+n}$.
>
> The remaining assumptions are not directly related to the specific application of our proposed algorithm but serve to support our general three-timescale stochastic approximation framework. This framework is a side-product of our paper and aims to ensure its generality and independent usage value. **Regarding the soundness of the convergence results**, we verify that our proposed algorithm (Algorithm 1) meets all the necessary assumptions on Pages 23-24 in the proof of convergence. Consequently, its associated ODE has a unique asymptotically stable equilibrium.
>
> These assumptions are standard and essential in the classical two-timescale stochastic approximation framework (refer to Section 6 in Borkar 2009), and we extend them to establish a three-timescale counterpart.
>
> ---
>
> **Q.** The reason why this DRO problem can be solved by the GD?
>
> **A.**  Although smoothness can contribute to faster convergence speeds when using GD, our DRRL problem is convex and Lipschitz with respect to the dual variable $\eta$, which is sufficient to establish asymptotic convergence.
>
> In our DRO problem (Equation 4), the gradient is almost defined on the entire real line, except when $\eta = \max_{a'}Q(s',a')$. In such cases, we can utilize the subgradient, and our algorithm will still function effectively. In fact, Duchi and Hongseok (2021) also consider the same DRO problem with Cressie-Read divergence of the $f$-family.
> Our DRRL problem is included in their problem if the $Q$ function is fixed.
> They apply gradient descent with backtracking Armijo line-searches to address the DRO problem for large datasets.
>
>  GD-based approaches are prevalent in past DRO studies, such as those by Namkoong Hongseok and John C. Duchi (2017), Jin, Jikai et al (2021), Qi, Qi et al (2021), Sinha, Aman et al (2017), and Blanchet, Jose et al (2022). Our work is closely related to Duchi and Hongseok (2021), focusing on the Cressie-Read divergence of the $f$-family. Our DRRL problem is included by their DRO problem when the $Q$ function is fixed. They use gradient descent with backtracking Armijo line-searches (Boyd, Stephen P. 2004) for large datasets (Duchi and Hongseok, 2021). Namkoong and John (2016) reformulate the DRO problem as a two-player game and develop a stochastic gradient-based solution combined with bandit learning. In the DRRL community, Wenhao Yang et al. (2023) propose a model-free DRRL algorithm using gradient descent. We believe GD-based DRRL solutions could be promising for efficient problem-solving, especially for online update requirements in RL applications.
>
> ---
>
> [1] Duchi, John C., and Hongseok Namkoong. "Learning models with uniform performance via distributionally robust optimization." The Annals of Statistics 49.3 (2021): 1378-1406.
>
> [2] Borkar, Vivek S. Stochastic approximation: a dynamical systems viewpoint. Vol. 48. Springer, 2009.
>
> [3] Namkoong, Hongseok, and John C. Duchi. "Stochastic gradient methods for distributionally robust optimization with f-divergences." Advances in neural information processing systems 29 (2016).
>
> [4] Namkoong, Hongseok, and John C. Duchi. "Variance-based regularization with convex objectives." Advances in neural information processing systems 30 (2017).
>
> [5] Yang, Wenhao, et al. "Avoiding model estimation in robust markov decision processes with a generative model." arXiv preprint arXiv:2302.01248 (2023).
>
> [6] Jin, Jikai, et al. "Non-convex distributionally robust optimization: Non-asymptotic analysis." Advances in Neural Information Processing Systems 34 (2021): 2771-2782.
>
> [7] Qi, Qi, et al. "An online method for a class of distributionally robust optimization with non-convex objectives." Advances in Neural Information Processing Systems 34 (2021): 10067-10080.
>
> [8] Sinha, Aman, et al. "Certifying some distributional robustness with principled adversarial training." arXiv preprint arXiv:1710.10571 (2017).
>
> [9] Boyd, Stephen P., and Lieven Vandenberghe. Convex optimization. Cambridge university press, 2004.
>
> [10] Blanchet, Jose, Karthyek Murthy, and Fan Zhang. "Optimal transport-based distributionally robust optimization: Structural properties and iterative schemes." Mathematics of Operations Research 47.2 (2022): 1500-1529.

---

> > ### Author Response · Authors · 2023-11-21
> > **Thanks for your insightful suggestions!**
> >
> > Dear reviewer,
> >
> >
> > Thank you once again for investing your valuable time in providing feedback on our paper. Your insightful suggestions have led to significant improvements in our work, and we look forward to possibly receiving more feedback from you. Since the discussion period between the author and reviewer is rapidly approaching its end, we kindly request you to review our responses. We firmly believe we address all your concerns, especially regarding our assumptions. In fact, we would like to highlight that, **in our application, we only need ONE mild assumption about the learning rate**, which is almost as mild as those non-robust $Q$-learning. Our gradient descent approach originated from existing DRO literature and is nature for DRRL problem. Additionally, we remain eager to engage in further discussion about any additional questions you may have.
> >
> >
> > Best regards,
> >
> > Authors

---

### Official Review · Reviewer_JJMJ · 2023-10-29

**Soundness:** 2 fair
**Presentation:** 1 poor
**Contribution:** 2 fair
**Rating:** 3
**Confidence:** 3

**Summary:**

In this paper, the authors proposed a distributionally robust variant of Q-learning aiming to solve distributionally robust MDP in an online fashion. Their algorithm utilizes a three-timescale stochastic approximation framework and possesses an almost surely convergence guarantee. They conduct thorough experiment illustrating the performance of their algorithm.

**Strengths:**

They extend the classical two-timescale stochastic approximation framework into the DRO problem, and design an online algorithm for DRRL. Comprehensive experiments are conducted to illustrate the performance.

**Weaknesses:**

The writing is problematic. I find the paper hard to follow. There are many notations/terms lack of definitions and several critical statements lack of discussion and explanation. Moreover, there are typos and factual errors in this paper, which make the results not credible. All in all, it's hard to verify the validity of this paper.

**Questions:**

1. What does the misspecified MDP in the paper mean?
2. Badrinath & Kalathil (2021) and Roy et al. (2017) didn't use R-contamination model. Their uncertainty set is a general one and covers yours. Both of them study the online setting and present asymptotic results. Why don't you compare with their works in experiment and compare their theoretical results with yours.
3. Why the first equation on page 5 is true?
4. On page 6, why keeping the learning speeds of $\eta$ and $Q$ different can stabilize the training process?
5. The three-timescale regime used in algorithm 1 is complicated and provided with limited explanations. Why and how it works?
6. In the proof, there are more than 10 assumptions and most of them don't have any explanation. Why do they hold in your case? Are they necessary? Are they reasonable in practical problems?

---

> ### Author Response · Authors · 2023-11-20
>
> We appreciate the reviewer's comments and efforts sincerely. We believe the following point-to-point response can address all the concerns and misunderstandings.
>
> ----
>
> **Q.** Could you please clarify the meaning of the misspecified MDP in the paper?
>
> **A.**  We apologize for not providing a formal definition of the misspecified MDP in the paper. In the context of distributionally robust reinforcement learning (DRRL), the main objective is to learn an optimal policy that is robust to unknown environmental changes. This becomes important when the transition model P and reward function r used during the training data collection differ from those in the test environment. In such cases, **we refer to the MDP environment (consisting of the transition model P and reward function r) used for training data collection as the misspecified MDP**. A detailed and rigorous definition can be found in Section 2.3. We appreciate your understanding and will ensure to provide clearer explanations in future work.
>
> -----------
>
> **Q.** Comparison with Roy et al. (2017) and Badrinath & Kalathil (2021).
>
> **A.** First, we would like to clarify that the uncertainty sets considered in Roy et al. (2017) and Badrinath & Kalathil (2021) **do not cover the ones in our work**, as they **do not require the transition model to be a valid probability transition** in the test environment. In fact, **their uncertainty set is equivalent to R-contamination set**. This leads to over-conservatism and significantly simplifies the theoretical difficulty and algorithmic design.
>
> In their works, Roy et al. (2017) and Badrinath & Kalathil (2021) use the following ambiguity set:
>
> $$
> P_i^a := \\{x+ p_i^a \lvert x\in U_{i}^a \\},
> $$
>
> where $p_i^a$ is the unknown state transition probability vector from the state $i\in \mathcal{X}$ to every other state in $\mathcal{X}$. The confidence region $U_i^a$ must be constrained to ensure $x+p_{i}^a$ lies within the probability simplex. However, they drop the requirement and consider a proxy confidence region.
> Recall the definition of the R-contamination model,
>
> $$
> P_s^a=\\{(1-R) p_s^a+R q \mid q \in \Delta_{|S|}\\}, s \in S, a \in A, \text { for some } 0 \leq R \leq 1.
> $$
>
> Note that  $(1-R)p_s^a+R q$ may also lie outside the probability simplex. Thus under the relaxation, the ambiguity set used in Roy et al. (2017) and Badrinath & Kalathil (2021) are equivalent to the R-contamination model.
> In contrast, our approach requires every element in the ambiguity set to lie within the probability simplex.
>
> This relaxation simplifies the algorithmic design and the convergence establishment. In particular, the $Q$-learning algorithm can be reformulated (see Equation (32) in Roy et al. (2017)) in terms of the operator $H$ as
> $$
> Q_t(i,a) = (1-\gamma_t) Q_{t-1}(i,a) + \gamma_t (HQ_t(i,a) + \eta_t(i,a)),
> $$
> where $\eta_t(i,a)$ is some martingale noise in the non-robust $Q$-learning.
>
> In the robust Q-learning approaches of Roy et al. (2017) and Badrinath & Kalathil (2021), they also benefit from the relaxation mentioned earlier, as the proxy uncertainty set in Equation 10 in Roy et al. (2017) **does not depend on the incoming sample $j$**. Despite having sufficient samples, their algorithm may still penalize some cases that will never occur, as their ambiguity set lacks distribution-awareness. Thus they mainly focus on ellipsoid and parallelepiped as concrete examples.
>
> In contrast, our approach maintains the worst-case distribution as a valid transition probability (referred to as distributionally aware) to prevent over-conservatism. As a result, $\eta_t(i,a)$ is not mean zero in our paper, as we require the transition to always fall within the probability complex. To address this challenge, our three-timescale algorithmic design, particularly the innermost loop and the second loop, along with the extended stochastic approximation framework, has been proposed.
>
> ---
>
>
> **Q.** Why the first equation on Page 5 is true?
>
> **A.** Recall that the definition of $c_k(\rho)$ defined in Lemma 3.1 as  $c_k(\rho) = (1+k(k-1)\rho)^{1/k}\ge 1$. Define
>
> $$
> f(\eta) = \eta-c_k(\rho)(\eta - \max_{a'}Q(s',a'))^+.
> $$
>
> Then the first order $f'(\eta) = 1 - c_k(\rho) \mathbb{1}\\{\eta \ge \max_{a'}Q(s',a')\\}$.
>
> We know $f'(\eta) = 1>0$ when $\eta <\max_{a'}Q(s',a')$ and $f'(\eta) = 0$ when $\eta \ge \max_{a'}Q(s',a')$.
>
> Thus $\max_{\eta\in \mathbb{R}}f(\eta) = f(\max_{a'}Q(s',a')) = \max_{a'}Q(s',a')$.
>
> Plug it into Equation 5 we have
> $$
> \begin{aligned}
> r(s,a)+\gamma \sup_{\eta\in \mathbb{R}}\\{\eta-c_k(\rho)(\eta - \max_{a'}Q(s',a'))\\}
> = r(s,a)+\gamma \{\max_{a'}Q(s',a')-c_k(\rho) \cdot 0\} = r(s,a) + \gamma \max_{a'}Q(s', a').
> \end{aligned}
> $$

---

> > ### Author Response · Authors · 2023-11-20
> >
> > **Q.** The connection between different learning speeds of $\eta$ and $Q$ with the stability of the training process.
> >
> > **A.**  Intuitively, $\eta(s,a)$ is independent of each state-action pair, whereas any state-action $Q$ value $Q(s,a)$ depends on other state-action pairs. Consequently, learning the optimal $Q$ value is more involved than learning the $\eta$ value. Due to this difference in their learnable nature, it is beneficial to set a relatively faster learning rate for $\eta$ to approximate the optimal value, while leaving the more challenging $Q$ function to be updated when $\eta$ is already accurate enough.
> >
> > From a theoretical standpoint, as suggested by the stochastic approximation framework (e.g., Borkar 2009), one of the main analysis tools for multiple-timescale Q-learning, setting two different learning rates (with specific conditions) allows the training process to be tracked as an ODE system. This enables a full description of the system's dynamics and guarantees convergence. The difference in learning rates is a result of this theoretical understanding.
> >
> > ----
> >
> > **Q.** Explain the three-timescale algorithm.
> >
> > **A.**  The DRRL problem can be reduced to finding a fixed point for Equation (5), which involves finding a $Q$ value that satisfies the equation:
> >
> > $$
> > Q(s, a)=r(s, a)+\gamma \sup _{\eta \in \mathbb{R}} \sigma_k(\max _{a^{\prime} \in \mathcal{A}} Q(\cdot, a^{\prime}),\eta).
> > $$
> >
> > Since there are two variables, $\eta$ and $Q$, in this equation, we update them alternatively to approach the fixed point:
> >
> > $$
> > \begin{aligned}
> > \eta_{n+1} &= \eta_n - \zeta_1 * \text{Gradient of } \eta_n,
> > \end{aligned}
> > $$
> > $$
> > Q_{n+1} =  r_n + \zeta_2 * \gamma \sigma_k(\max_{a'\in \mathcal{A}} Q(\cdot, a'), \eta_n).
> > $$
> >
> > To obtain a better estimator for the gradient of $\eta_n$, we introduce two auxiliary variables, $Z_1$ and $Z_2$, to approximate the true ingredients in the gradient of $\eta_n$.
> >
> > Combining all these components and arranging proper learning speeds for them, we have the following algorithm:
> >
> > $$
> > Z_{n+1, 1} = \text{Update}(Z_{n,1})
> > $$
> >
> > $$
> > Z_{n+1, 2} = \text{Update}(Z_{n,2})
> > $$
> >
> > $$
> > \eta_{n+1} = \eta_n - \zeta_1 * \text{Gradient of } \eta_n
> > $$
> >
> > $$
> > Q_{n+1} =  r_n + \zeta_2 * \gamma \sigma_k(\max_{a'\in \mathcal{A}} Q(\cdot, a'), \eta_n).
> > $$
> >
> > The complexity in Algorithm 1 primarily arises from the explicit formula of the dual problem of the Cressie-Read divergence of the $f$ family and the gradient of $\eta_n$.
> >
> > ------
> >
> > **Q.** Too many assumptions.
> >
> > **A.** We apologize for the confusing presentation of the results, particularly the organization of the assumptions.
> >
> > For our main theoretical result (Theorem 3.3), which concerns the asymptotic convergence of the DRRL algorithm, **we only require ONE assumption about the learning rate (Assumption C.10)**. This assumption can easily be satisfied by some common choices, such as $\zeta_1(n) = \frac{1}{1+n^{0.6}}$, $\zeta_2(n) = \frac{1}{1+n^{0.8}}$, and $\zeta_3(n) = \frac{1}{1+n}$, and thus is almost as mild as the nonrobust $Q$ learning algorithm.
> >
> > The remaining assumptions are not directly related to the specific application of our proposed algorithm but are included to support our general three-timescale stochastic approximation framework, which serves as a side-product in our paper. The purpose of including these assumptions is to ensure the framework is general enough to have independent usage value. These assumptions are standard and necessary in the classical two-timescale stochastic approximation framework (refer to Section 6 in Borkar 2009).
> >
> > We confirm that our proposed algorithm (Algorithm 1) satisfies all the necessary assumptions on Pages 23-24 in the proof of convergence.
> >
> > ---
> >
> > [1] Borkar, Vivek S. Stochastic approximation: a dynamical systems viewpoint. Vol. 48. Springer, 2009.

---

> > > ### Author Response · Authors · 2023-11-21
> > > **Thanks for your insightful suggestions!**
> > >
> > > Dear reviewer,
> > >
> > > Thank you once again for investing your valuable time in providing feedback on our paper. Your insightful suggestions have led to significant improvements in our work, and we look forward to possibly receiving more feedback from you. Since the discussion period between the author and reviewer is rapidly approaching its end, we kindly request you to review our responses. We firmly believe we address all your concerns. Additionally, we remain eager to engage in further discussion about any additional questions you may have, especially those concerning the "factual errors" you found in our paper. We are more than happy to offer further explanation and illustration of our ideas to ensure clarity.
> > >
> > > Best,
> > >
> > > Authors

---

> > ### Comment · Reviewer_JJMJ · 2023-12-05
> >
> > I would like to thank the authors for their response. However, I am still concerned about the authors' explanation of some of my questions.
> >
> > **1.** The authors state that
> > >Note that $(1-R)p_s^a + Rq$ may also lie outside the probability simplex.
> >
> > This is not true. Since $p_s^a$ is the nominal transition probability corresponding to $(s,a)$, it must be in the probability simplex, and $q$ is also in the probability simplex by definition. Note that $R$ is a scalar satisfying $0\leq R\leq 1$. Denote $\tilde{p}=(1-R)p_s^a + Rq$, then $\tilde{p}$  is a mixture distribution of $p_s^a$ and $q$ satisfying (1) $0\leq \tilde{p}_i \leq 1$, and (2)$ {\tilde {p}}_1 + \cdots + {\tilde {p}}_d=1$ , thus $\tilde{p}$ must lie inside the probability simplex.
> >
> > **2.** The authors' logic on arguing ``Roy et al. (2017) and Badrinath & Kalathil (2021)'s uncertainty set is equivalent to R-contamination set`` is based on the following argument:
> > >(1) elements in Roy et al. (2017) and Badrinath & Kalathil (2021)'s uncertainty set may lie outside the probability simplex; (2) elements in the R-contamination model may also lie outside the probability simplex.
> >
> > Based on these, they draw the conclusion
> > >Thus under the relaxation, the ambiguity set used in Roy et al. (2017) and Badrinath & Kalathil (2021) are equivalent to the R-contamination model.
> >
> > I cannot understand this logic. In order to state two sets are equivalent, the authors need to show that the two sets contain exactly the same elements.
> >
> > **3.** The authors' logic on arguing ``Roy et al. (2017) and Badrinath & Kalathil (2021) do not cover the ones in our work`` is based on
> > >Roy et al. (2017) and Badrinath & Kalathil (2021)'s uncertainty set contains not only elements in the probability simplex, but also elements lie outside the probability simplex, while their uncertainty set only contains probability distributions.
> >
> > Isn't this just saying that Roy et al. (2017) and Badrinath & Kalathil (2021)'s uncertainty set is larger than yours, which means that it probably contains yours? In order to say Roy et al. (2017) and Badrinath & Kalathil (2021)'s uncertainty set cannot cover yours, the author need to show that there exist at least one element in your uncertainty set that is not contained in Roy et al. (2017) and Badrinath & Kalathil (2021)'s uncertainty set.
> >
> > Due to factual errors in the authors' response and the omission of my comment on empirical comparison with Roy et al. (2017) and Badrinath & Kalathil (2021)'s online method, I am skeptical about the overall correctness of this paper, particularly the proof section that I couldn't scrutinize in detail. Therefore, I will maintain my original score.

---

### Official Review · Reviewer_DkEx · 2023-10-29

**Soundness:** 3 good
**Presentation:** 2 fair
**Contribution:** 2 fair
**Rating:** 6
**Confidence:** 5

**Summary:**

This paper considers a distributionally robust RL problem. A model-free online TD Q-learning type method is developed to find the robust Q values without relying on a simulator. Specifically, a three-timescale framework is introduced to approximate the robust Bellman equation and asymptotic analysis is provided. The main algorithm is validated through a tabular RL example; a more practical DQN type algorithm is introduced to handle more complicated examples.

**Strengths:**

- This paper considers distributionally robust MDP using f-divergence as the uncertainty set, which is novel.
- The motivations of not using SAA to approximate robust bellman equation are clear.
- The reasons of not using multilevel Monte-Carlo method are clear.

**Weaknesses:**

- I guess the paper was written in parallel with the Panaganti 2022 (Robust offline) paper. However, as the Panaganti 2022 paper is published for over 6 months, it is better to compare and discuss the difference choices of uncertainty sets and problem settings. The authors claim that this paper is the first model-free DR RL paper in the literature, which is not true as the Panaganti 2022 is also model-free. To some extent, their paper considers a harder offline problem while this paper considers an online version.
- I don't think this is an issue when evaluating the contribution of this paper. However, it is better to state the contribution according to the latest literature (indeed, RFQI was mentioned in the experiment) and explicitly comment on the differences.
- The proposed algorithm 1 only works in the tabular setting. This is fine. The authors introduce a more practical algorithm using DQN type of learning scheme, which is provided in algorithm 4 in the Appendix. However, algorithm 4 looks almost the same as algorithm 1. Is this paper an older version? If yes, please provide the full algorithm 4 description in the future. More importantly, I believe algorithm 4 could potentially have a greater impact on DRRL problems; its connection and modifications with algorithm 1 could be further explained in details in the main paper.

**Questions:**

- In my opinion, the combination of online RL with DRO is a bit weird. It makes more sense to study offline RL with DRO. The question is that why would you prefer to learn a robust policy in the testing environment, i.e., this robust policy is not optimal in the testing environment. Is it because there are some technical challenges when applying f-diverngence to offline dataset, e.g., when no e-greedy policy is allowed?
- Same to the weakness, please compare with Panaganti 2022 in details and explicitly.

---

> ### Author Response · Authors · 2023-11-20
>
> We thank gratefully the reviewer for various valuable suggestions and the praise of our interesting findings and clear writing! The points you raised are explained in the following.
>
> ---
>
> **Q.** Comparison with Panaganti et al. 2022.
>
> **A.**  We appreciate your feedback and agree that our initial claim in the abstract may have been overstated.  We will make the necessary corrections to our claim.
>
> We acknowledge the valuable contribution of the RFQI algorithm proposed by Panaganti et al. in 2022 and their novel work in addressing the DR offline RL problem with both theoretical guarantees and practical applications. Our primary focus is on the single trajectory scheme, which presents unique challenges compared to other papers, including Panaganti et al. 2022.
>
> At the time of completion of our paper, the existing literature on model-free DRRL was limited, with the closest works being Panaganti et al. 2022 and Liu et al. 2022. We have already provided a thorough comparison of our algorithm with Liu et al. 2022 in the main text.
>
>  We would like to kindly point out the primary distinctions between our proposed algorithm and Panaganti's approach:
>
>
> 1. Ambiguity Set: RFQI concentrates on the **total variation distance**, while we consider the Creese-Read family of $f$-divergence which can cover several commonly used divergence.
> 2. Technical Assumptions: RFQI heavily relies on the **fail-state** assumption, which assumes the existence of a state that yields a 0 reward regardless of the chosen action and must return to itself once entered. Such assumption aids to bypass the nonlinearity in the dual problem of the DRO problem with TV distance, i.e., the $\inf_{s''}V(s'')$ part in Equation 4 in Panaganti et al. 2022, and significantly simplify the estimation. This assumption may not be applicable in certain cases, such as infinite-horizon problems. Instead, the dual problem of the Creese-Read family of $f$-divergence is nonliear with respect the expectation part and we don't impose assumption to remove the nonlinearity. Instead we propose our multi-timescale algorithmic design to address it directly.
> 4. Data Collection Requirement: While RFQI aims to learn an optimal DR RL policy using a pre-collected offline dataset, our algorithm offers greater flexibility by allowing data to be fed in a single trajectory manner, which includes the pre-collected dataset setting.
> 5. Algorithmic Design: RFQI is based on the batch data scheme and develops their algorithm from value iteration with function approximation, alternating between updating the dual variable and the $Q$ value to ensure satisfactory optimization before updating the other variable. Conversely, our algorithm is grounded in the $Q$-learning approach and incrementally updates both the dual variable and the $Q$ value function simultaneously upon each data arrival, albeit with distinct learning rates.
>
>
>    Our algorithm employs the Cressie-Read family of $f$-divergence and can accommodate several common divergences, including KL and $\chi^2$ divergence. The dual problem under this family is highly nonlinear with respect to the expectation, which is addressed by our multiple-timescale algorithmic design. Additionally, we do not impose extra assumptions for our chosen ambiguity set.
>
> |  | RFQI | DRQ (Ours) |
> | --- | --- | --- |
> | Ambiguity Set | Total Variation Distance | Cressie-Read family of f-divergence |
> | Assumption for Ambiguity Set | Fail-State | No assumptions specific for ambiguity set |
> | Data Collection | A batch of pre-collected data | Allows even single-trajectory data |
> | Algorithmic Design | Value Iteration with alternative update between dual variable and Q | Q-learning with incremental and simultaneous update for dual variable and Q |

---

> > ### Comment · Reviewer_DkEx · 2023-11-21
> >
> > Thanks. I guess RFQI might only work under TV distance as they want to use least square to find the Q function in the offline setting. If KL is used, as you said, it is nonlinear w.r.t the expectation and LS does not work.
> > Do you think it is possible to apply the type of method you developed to the offline setting?

---

> > > ### Author Response · Authors · 2023-11-22
> > >
> > > Yes, our multi-timescale update scheme is a generic approach to obtain satisfying estimations under nonlinearity, and thus, it can be developed for the offline setting.

---

> ### Author Response · Authors · 2023-11-20
>
> **Q.** Clarification about Algorithm 4.
>
> **A.** We are sincerely sorry that mistakenly attaching the Algorithm 4 to the appendix, which is the same as Algorithm 1. **The more practical algorithm using DQN type of learning scheme is summarized into Algorithm 2**.
>
> In particular, in Algorithm 2, we utilize the Deep Q-Network (DQN) architecture and neural networks as functional approximators for the dual variable and Q function in place of the tabular function in Algorithm 1. To improve training stability, we introduce target networks that are updated at a slower rate. We adopt a two-timescale update approach, which is employed to minimize the Bellman error for the Q network and maximize the DR Q value for the dual variable network. Even though this may introduce bias in the dual variable's convergence, resulting in a lower target value for the Q network, this approach serves as a robust update strategy for our DRRL problem, with further discussion in Appendix B.3. Moreover, Algorithm 1 updates on each data arrival, while the practical implementation uses a batch scheme, training the neural networks on a resampled subset of samples with a replay buffer retaining all previous samples.
> We will move its connection and modifications with algorithm 1 to the main text.
>
>
>
> -------
>
>
> **Q.** The motivation of the combination of online RL with DRO.
>
>  **A.** We appreciate the reviewer's suggestion that combining DRO with offline RL would have more direct application value. One potential application of our online DRRL algorithm is in situations where the test environment's transition or reward functions change incrementally over time. In such cases, a promising algorithm should continuously learn a robust policy to counter potential upcoming perturbations.
>
> Our paper aligns with the model-free Robust RL literature, such as works by Dong, Jing, et al. (2022), Badrinath, et al. (2021), and Wang, Yue, and Shaofeng Zou (2021). These works present model-free algorithms capable of learning optimal robust policies without the need for a simulator or a pre-collected dataset. Our novel analysis of the Cressie-Read family of $f$-divergence and the multi-timescale algorithmic design are generic and can be easily integrated with other non-robust offline RL algorithms, such as the CQL algorithm (Kumar, Aviral, et al., 2020). We chose to present our method using online learning to demonstrate its value under the most stringent update requirements.
>
> ---
>
> [1] Dong, Jing, et al. "Online policy optimization for robust MDP." arXiv preprint arXiv:2209.13841 (2022).
>
> [2] Badrinath, Kishan Panaganti, and Dileep Kalathil. "Robust reinforcement learning using least squares policy iteration with provable performance guarantees." International Conference on Machine Learning. PMLR, 2021.
>
> [3] Wang, Yue, and Shaofeng Zou. "Online robust reinforcement learning with model uncertainty." Advances in Neural Information Processing Systems 34 (2021): 7193-7206.
>
> [4] Kumar, Aviral, et al. "Conservative q-learning for offline reinforcement learning." Advances in Neural Information Processing Systems 33 (2020): 1179-1191.

---

> > ### Author Response · Authors · 2023-11-21
> > **Thanks for your insightful suggestions!**
> >
> > Dear reviewer,
> >
> > Thank you once again for investing your valuable time in providing feedback on our paper. Your insightful suggestions have led to significant improvements in our work, and we look forward to possibly receiving more feedback from you. Since the discussion period between the author and reviewer is rapidly approaching its end, we kindly request you to review our responses to ensure that we have addressed all of your concerns. Also, we remain eager to engage in further discussion about any additional questions you may have.
> >
> > Best,
> >
> > Authors

---

> > > ### Comment · Reviewer_DkEx · 2023-11-21
> > >
> > > I think the authors address my concerns. I raised my score to 6.
> > >
> > > Some of the strengths and improvements compared to previous papers are due to the online setting (continuous explorations, two-timescale stochastic approximation). Besides, I am not convinced by the promising applications of applying DRO to online RL problems.

---

> > > > ### Author Response · Authors · 2023-11-22
> > > >
> > > > Thank you so much for recognizing our strengths! Even though applying DRO to online RL currently has vague value in the application, we believe our proposed method can motivate further exploration in the DRRL problem with the general $f$-divergence.

---

> > > > ### Author Response · Authors · 2023-11-23
> > > > **Further clarifications on applicability**
> > > >
> > > > We would like to further clarify the applicability issue. Sim2Real is an important class of problems where our algorithms and results would apply. Given a high-fidelity simulator, modern RL applications would often train/learn a policy therein before deploying it in a real environment. Such industrial-grade simulators are highly complex (pre-packaged by manufacturers) and training must follow a single trajectory until the end before starting another episode of training. In such cases, the (unknown) testing environment is often a real environment, and has a sim2real (simulator-to-reality) gap that the RL agent must accomodate. As such,  instead of deploying the optimal policy for the simulator, we propose to deploy the optimal distributionally robust policy for the simulator.

---

### Official Review · Reviewer_ogVe · 2023-10-31

**Soundness:** 3 good
**Presentation:** 3 good
**Contribution:** 2 fair
**Rating:** 8
**Confidence:** 4

**Summary:**

This work designs model-free algorithms for distributionally robust RL problems in a sample-efficient manner. It proposed a three-time scale algorithm that solves a class of robust RL problems using the uncertainty set constructed by the Cressie-Read family of f-divergence. A theoretical asymptotic guarantee has been provided. Moreover, this work conducted experiments to evaluate the performance and sample efficiency of this work.

**Strengths:**

1. It targets an interesting problem: design a model-free algorithm for distributionally robust RL problems.
2. A three-timescale algorithm has been proposed that enjoys an asymptotic guarantee and practical sample efficiency.
3. The introduction of the algorithm is clear and easy to follow.

**Weaknesses:**

1. For the experiments in Figure 5. It seems the proposed algorithm DDQR has a very similar performance compared to the existing robust algorithm SR-DQN. It will be helpful to add more discussion about this.
2. As mentioned in the algorithm, the three-timescale serves as a key role in the algorithm to ensure convergence. So it will be better to introduce what is the three learning rates that the practical algorithm uses. And ablation study using different learning rates will give a message about whether the algorithm is sensitive to the learning rate.

**Questions:**

1. As the update of $\eta$ is independent for each $(s,a)$ pair, will the proposed algorithm works for $s$-rectangular cases?

---

> ### Author Response · Authors · 2023-11-20
>
> Thank you for your detailed reading and valuable comments. Please find our response to the comments below:
>
> ---
>
> **Q.** Disscussion about the similar performance of the proposed DDQR algorithm with the existing robust algorithm SR-DQN.
>
> **A.** The SR-DQN algorithm demonstrates comparable performance to our proposed DDQR method, primarily when environmental parameters are perturbed (e.g., FMP perturbation in the Cartpole environment and EPP perturbation in the LunarLander environment). In these perturbations, even if the force magnitude (FMP) or engine power (EPP) is altered in the test environment, the transition remains almost deterministic, meaning only one state transitions when an action is taken under the current state. Consequently, these perturbations maintain a similar level of randomness as the original environment, which might not substantially degrade the learned policy. As a result, the soft-robustness principle employed in SR-DQN may be sufficient to manage the perturbation, leading to similar performance as our proposed algorithm.
>
> However, when it comes to action perturbations in both the Cartpole and LunarLander environments, multiple states can be transitioned as different actions are deployed in the environment due to the perturbation. This increases the randomness level, and SR-DQN is notably outperformed by our DDQR algorithm, as it does not provide adequate robustness.
>
> It is worth mentioning that SR-DQN does perform well in certain situations, as demonstrated in the Cartpole experiment by Panaganti et al. (2022).
>
> ---
>
> **Q.** The potential of the proposed algorithm to work in s-rectangular setting.
>
> **A.** We agree with the reviewer's insight that our proposed algorithm can work in s-rectangular setting by tracking the optimal dual variable for each state, rather then for each state-action pair in the current sa-rectangular setting.
>
> ---
>
> [1] Panaganti, Kishan, et al. "Robust reinforcement learning using offline data." Advances in neural information processing systems 35 (2022): 32211-32224.

---

> > ### Author Response · Authors · 2023-11-20
> >
> > **Q.** More details about the three learning rates in the practical algorithm and conduct an ablation study to assess its sensitivity to learning rates.
> >
> > **A.** Our practical algorithm is updated in a two-timescale as the neural network already introduce a nonnegligible bias in the estimation, we allow the update of the dual variable from a biased gradient estimator and thus drop the usage of another timescale. We choose $2.5e^{-4}$ as the learning rate of the $Q$ network and $2.5e^{-3}$ as the learning rate of the dual variable and use the Adam optimizer to search for the optimal values.
> >
> > We conduct an additional ablation study to assess the robustness of our DDRQ. Utilizing the same experimental setups employed in the CartPole and LunarLander environments, as described in Section 4.2, we vary the learning rates for both the $Q$ network and the dual variable network. The summarized experimental results are presented in the following tables. For the CartPole environment, we test five groups of learning rates: $\{(10^{-4}, 10^{-3}), (2.5\times 10^{-4}, 2.5\times 10^{-3}), (5\times 10^{-4}, 5\times 10^{-3}), (10^{-4}, 10^{-2}), (10^{-3}, 10^{-2})\}$. Among these, the pair $(2.5\times 10^{-4}, 2.5\times 10^{-3})$ was previously presented in Section 4.2, and we rescale it to $(10^{-4}, 10^{-3})$, $(5\times 10^{-4}, 5\times 10^{-3})$, and $(10^{-3}, 10^{-2})$. While maintaining the same learning rates for the $Q$ network and the dual network, we vary their relative ratio by examining the pair $(10^{-4}, 10^{-2})$. Our experimental results suggest that our DDRQ exhibits relatively stable performance under different choices of learning rates, even when their ratio is altered. However, a larger learning rate in $Q$ network may render poorer performance, e.g., $(10^{-4}, 10^{-2})$ and $(10^{-3}, 10^{-2})$ pairs in Cartpole Action Perturbation experiment (Table 2). Also altering the ratio of the learning speeds may affect the performance, see $(10^{-4}, 10^{-2})$ in Cartpole FMP perturbation experiment (Table 1).
> > It is important to note that for the more complex LunarLander environment, the pair $(10^{-4}, 10^{-3})$ is too small for our algorithm to converge to an optimal policy, while $(10^{-3}, 10^{-2})$ is too large for convergence within our two-timescale approximation scheme. Consequently, we only report the remaining pairs in Tables 3 and 4.
> >
> >   | % Change | Learning Rates | $(10^{-4}, 10^{-3})$ | $(2.5\times 10^{-4}, 2.5\times 10^{-3})$ | $(5\times 10^{-4}, 5\times 10^{-3})$ | $(10^{-4}, 10^{-2})$ | $(10^{-3}, 10^{-2})$ |
> > |---|---|---|---|---|---|---|
> > | -80 |  | 196.8 (± 2.0) | 196.6 (± 4.9) | 195.9(± 2.4) | 168.9 (± 70.1) | 197.5 (± 5.13) |
> > | -70 |  | 197.7 (± 2.1) | 198.5 (± 2.1) | 198.3(± 5.8) | 171.8 (± 44.6) | 196.7 (± 2.13) |
> > | -60 |  | 200.0 (± 0.0) | 198.8 (± 1.9) | 200.0(± 0.0) | 171.2 (± 84.2) | 200.0 (± 0.0) |
> > | -50 |  | 200.0 (± 0.0) | 200.0 (± 0.0) | 200.0(± 0.0) | 178.0 (± 46.1) | 200.0 (± 0.0) |
> > | -40 |  | 200.0 (± 0.0) | 199.6 (± 0.9) | 200.0(± 0.0) | 178.8 (± 52.2) | 200.0 (± 0.0) |
> > | -30 |  | 200.0 (± 0.0) | 200.0 (± 0.0) | 200.0(± 0.0) | 184.7 (± 31.2) | 200.0 (± 0.0) |
> > | -20 |  | 199.8 (± 0.8) | 200.0 (± 0.0) | 200.0(± 0.0) | 189.1 (± 22.1) | 200.0 (± 0.0) |
> >
> > Table 1: **Cartpole FMP Perturbation with Varied Learning Rates:** The first element in the learning rate tuple pertains to the Q network, while the second corresponds to the dual variable network. Each experiment is replicated 100 times, and the mean and standard deviation of the returns are reported.
> >
> >
> > | % Random | Learning Rates | $(10^{-4}, 10^{-3})$ | $(2.5\times 10^{-4}, 2.5\times 10^{-3})$ | $(5\times 10^{-4}, 5\times 10^{-3})$ | $(10^{-4}, 10^{-2})$ | $(10^{-3}, 10^{-2})$ |
> > |---|---|---|---|---|---|---|
> > | 0 |  | 200.0 (± 0.0) | 200.0 (± 0.0) | 200.0(± 0.0) | 196.6 (± 3.7) | 200.0 (± 0.0) |
> > | 10 |  | 200.0 (± 0.0) | 200.0 (± 0.0) | 200.0(± 0.0) | 157.5 (± 35.3) | 200.0 (± 0.0) |
> > | 20 |  | 194.9 (± 3.8) | 200.0 (± 0.0) | 196.4(± 5.9) | 166.7 (± 26.6) | 192.1 (± 5.8) |
> > | 30 |  | 187.3 (± 19.1) | 199.3 (± 1.8) | 197.5(± 2.4) | 149.4 (± 43.7) | 192.8 (± 6.7) |
> > | 40 |  | 164.4 (± 25.1) | 180.7 (± 17.8) | 180.5(± 21.38) | 124.7 (± 39.3) | 151.7 (± 35.3) |
> > | 50 |  | 121.7 (± 46.9) | 131.8 (± 53.8) | 133.1(± 41.4) | 72.3 (± 44.2) | 124.0 (± 73.4) |
> > | 60 |  | 88.3 (± 73.8) | 84.7 (± 61.2) | 81.4(± 68.7) | 43.3 (± 43.7) | 67.3 (± 41.9) |
> > | 70 |  | 65.3 (± 54.9) | 66.2 (± 65.1) | 59.4(± 57.4) | 37.5 (± 48.2) | 57.7 (± 49.8) |
> >
> > Table 2: **Cartpole Action Perturbation with Varied Learning Rates:** The first element in the learning rate tuple pertains to the Q network, while the second corresponds to the dual variable network. Each experiment is replicated 100 times, and the mean and standard deviation of the returns are reported.

---

> > > ### Author Response · Authors · 2023-11-20
> > >
> > > | % Change | Learning Rates | $(10^{-4}, 10^{-2})$ | $(2.5\times 10^{-4}, 2.5\times 10^{-3})$ | $(5\times 10^{-4}, 5\times 10^{-3})$ |
> > > |---|---|---|---|---|
> > > | -60 |  | -82.8 (± 48.5) | -74.9 (± 42.5) | -82.3(± 34.8) |
> > > | -50 |  | -29.4 (± 38.7) | -29.1 (± 32.4) | -23.1(± 27.7) |
> > > | -40 |  | 19.1 (± 35.5) | 12.9 (± 42.6) | 17.2(± 26.2) |
> > > | -30 |  | 141.3 (± 114.0) | 152.1 (± 61.4) | 161.7(± 121) |
> > > | -20 |  | 191.5 (± 94.4) | 236.6 (± 84.5) | 231.9(± 45.8) |
> > > | -10 |  | 247.6 (± 38.4) | 262.9 (± 40.1) | 222.2 (± 31.8) |
> > > | 0 |  | 269.4 (± 27.4) | 257.7 (± 79.8) | 243.6(± 29.9)|
> > >
> > > Table 3: **LunarLander Engines Power Perturbation with Varied Learning Rates:** The first element in the learning rate tuple pertains to the Q network, while the second corresponds to the dual variable network. Each experiment is replicated 100 times, and the mean and standard deviation of the returns are reported.
> > >
> > >
> > > | % Random | Learning Rates | $(10^{-4}, 10^{-2})$ | $(2.5\times 10^{-4}, 2.5\times 10^{-3})$ | $(5\times 10^{-4}, 5\times 10^{-3})$ |
> > > |---|---|---|---|---|
> > > | 0 |  | 259.8 (± 73.9) | 257.0 (± 79.8) | 217.6(± 53.1) |
> > > | 10 |  | 231.0 (± 44.6) | 242.3 (± 40.1) | 201.6(± 53.2) |
> > > | 20 |  | 181.7 (± 106.5) | 214.8 (± 84.5) | 191.8(± 54.2) |
> > > | 30 |  | 135.8 (± 75.3) | 130.5 (± 61.4) | 113.0(± 68.4) |
> > > | 40 |  | 39.8 (± 39.4) | 41.7 (± 42.6) | 33.4(± 27.2) |
> > > | 50 |  | -0.1 (± 50.6) | -2.9 (± 32.4) | 8.7 (± 36.17) |
> > > | 60 |  | -33.8 (± 39.1) | -10.5 (± 42.5) | -15.9(± 32.2)|
> > >
> > > Table 4: **LunarLander Action Perturbation with Varied Learning Rates:** The first element in the learning rate tuple pertains to the Q network, while the second corresponds to the dual variable network. Each experiment is replicated 100 times, and the mean and standard deviation of the returns are reported.

---

> > > > ### Author Response · Authors · 2023-11-21
> > > > **Thanks for your insightful suggestions!**
> > > >
> > > > Dear reviewer,
> > > >
> > > > Thank you once again for investing your valuable time in providing feedback on our paper. Your insightful suggestions have led to significant improvements in our work, and we look forward to possibly receiving more feedback from you. Since the discussion period between the author and reviewer is rapidly approaching its end, we kindly request you to review our responses to ensure that we have addressed all of your concerns. Also, we remain eager to engage in further discussion about any additional questions you may have.
> > > >
> > > > Best,
> > > >
> > > > Authors

---

> > > > > ### Comment · Reviewer_ogVe · 2023-11-22
> > > > > **Response to the author**
> > > > >
> > > > > Thanks for providing the additional ablation study for the learning rate. The answers address the reviewer's concerns and the reviewer would like to keep the positive scores for this work.

---

### Official Review · Reviewer_UFU7 · 2023-11-01

**Soundness:** 3 good
**Presentation:** 3 good
**Contribution:** 3 good
**Rating:** 6
**Confidence:** 2

**Summary:**

The paper proposes a novel distributionally robust reinforcement learning (DRRL) algorithm which is model-free and uses single trajectories to update Q function estimates and compute the robust policy. The approach consists of a multi-scale approximation scheme that utilizes existing stochastic approximation results. Asymptotic convergence is proven. Moreover, the method is evaluated experimentally on two tabular environments and a DQN-based implementation is tested on larger (and widely used) control tasks.

**Strengths:**

- Distributionally robust reinforcement learning is a relevant and active area of research. While a few methods have been already proposed, the paper is novel in that it considers a model-free setting, without assuming access to a simulator but only to single trajectory data. I believe this is significantly more practical and makes a good contribution.

- A reasonable amount of experiments illustrate the features of the proposed approach, compared to existing DRRL baselines.

- The paper is nicely written and the results look sound, although i could not verify their proofs.

**Weaknesses:**

- Only asymptotic convergence is proven, and no sample-complexity guarantees. Do the authors have a guess on how these may compare with previous work, e.g. (Panaganti et al. 2022)?

- In Section 4.2, a practical implementation of DQR is utilized for the experiments. How is this implemented? I would be nice to discuss such implementation in the main text.

**Questions:**

See weaknesses.

---

> ### Author Response · Authors · 2023-11-20
>
> Thank you for your review and positive comments on our paper. Please find our response to the comments below:
>
> ------
>
> **Q. Comparison of the sample-complexity of the proposed algorithm with Panaganti et al. 2022.**
>
> **A.**  Based on the current literature, a reasonable guess of the sample complexity of our proposed algorithm might be $O(n^{-1/3})$, which may not be as efficient as the $O(n^{-1/2})$ in Panaganti et al. 2022. Our algorithm is based on the multi-timescale nonlinear stochastic approximation method, which, to our knowledge, has not been considered for finite sample efficiency in existing literature. While previous works have mostly focused on the linear case for the two-timescale scenario, there is limited information available for the nonlinear counterpart.
>
> Broadly speaking, for linear stochastic approximation, a notable result in Dalal et al. 2018 suggests that the sample efficiency can be at most $O(n^{-1/3})$, where $n$ represents the sample size. Subsequently, Kaledin Maxim et al. 2020 and Dalal Gal et al. 2020 improved this rate to $O(n^{-1/2})$ by further exploiting the linear structure. In the case of nonlinear two-timescale stochastic approximation, Doan Thinh T has established a $O(n^{-1/3})$ sample efficiency guarantee. Given the current progress in understanding two-timescale nonlinear stochastic approximation, we anticipate that the optimal sample efficiency we can achieve is $O(n^{-1/3})$.
>
> -------
>
> **Q: Discussion about the practical implementation of DQR algorithm.**
>
> **A:** We have deferred the discussion of the specific implementation to Appendix B.3 and summarized into Algorithm 2. In this answer, we restate the implementation details, particularly the differences from the DRQ algorithm (Algorithm 1), and will move the main part of the discussion to the main text.
>
> Instead of the tabular function in Algorithm 1, we adopt the Deep Q-Network (DQN) architecture (Mnih et al., 2015) and choose neural networks as functional approximators for the dual variable and the Q function. To enhance training stability, we introduce another set of neural networks as the corresponding target networks, which are updated at a slower rate. Due to the existence of approximation error, we adopt a two-timescale update approach: our Q network aims to minimize the Bellman error, while the dual variable $\eta$ network strives to maximize the DR Q value. The two-timescale update approach could introduce bias in the convergence of the dual variable, and thus the dual variable η may not be the optimal dual variable for the primal problem. Given the primal-dual structure of this DR problem, this could render an even lower target value for the Q network to learn. This approach can be understood as a robust update strategy for our original DRRL problem. More discussion about the validity of this update strategy can be found in Appendix B.3.
>
> Another significant difference is in the update scheme. While in Algorithm 1, we update on each data arrival, in the practical implementation, the neural networks are updated in a batch scheme, i.e., trained on a resampled subset of samples with a replay buffer keeping all the previous samples.
>
> ------
>
> [1] Mnih, Volodymyr, et al. "Human-level control through deep reinforcement learning." nature 518.7540 (2015): 529-533.
>
> [2] Dalal, Gal, et al. "Finite sample analysis of two-timescale stochastic approximation with applications to reinforcement learning." Conference On Learning Theory. PMLR, 2018.
>
> [3] Panaganti, Kishan, et al. "Robust reinforcement learning using offline data." Advances in neural information processing systems 35 (2022): 32211-32224.
>
> [4] Kaledin, Maxim, et al. "Finite time analysis of linear two-timescale stochastic approximation with Markovian noise." Conference on Learning Theory. PMLR, 2020.
>
> [5] Dalal, Gal, Balazs Szorenyi, and Gugan Thoppe. "A tale of two-timescale reinforcement learning with the tightest finite-time bound." Proceedings of the AAAI Conference on Artificial Intelligence. Vol. 34. No. 04. 2020.
>
> [6] Doan, Thinh T. "Nonlinear two-time-scale stochastic approximation convergence and finite-time performance." IEEE Transactions on Automatic Control (2022).

---

> > ### Author Response · Authors · 2023-11-21
> > **Thanks for your insightful suggestions!**
> >
> > Dear reviewer,
> >
> > Thank you once again for investing your valuable time in providing feedback on our paper. Your insightful suggestions have led to significant improvements in our work, and we look forward to possibly receiving more feedback from you. Since the discussion period between the author and reviewer is rapidly approaching its end, we kindly request you to review our responses to ensure that we have addressed all of your concerns. Also, we remain eager to engage in further discussion about any additional questions you may have.
> >
> > Best,
> >
> > Authors

---

### Author Response · Authors · 2023-11-22
**Summary of the Rebuttal by Authors**

We express our gratitude to the reviewers for their meticulous examination of the paper and their insightful feedback, which we highly value. In summary, we have addressed the following concerns raised by the reviewers with adequate evidence:

1. **Comparison with Previous Literature:**
   - Conducted a thorough comparison with Panaganti et al. 2022, emphasizing the flexibility of our Creese-Read family of $f$-divergence, mild assumptions, relaxed data collection requirements, and novel algorithmic design.
   - Highlighted distinctions from Roy et al. (2017) and Badrinath & Kalathil (2021), pointing out that the ambiguity set they consider is, in fact, an R-contamination model after their relaxation.
   - Established connections with previous DRO literature that uses a gradient-descent approach, aiding in understanding the correctness and efficiency of our method. Also, provided an example from recent DRRL literature employing a similar GD-based approach, highlighting its potential in the DRRL domain.

2. **Practical Implementation:**
   - Outlined details about our practical implementation (DDRQ algorithm) for the reviewers to evaluate its practical value.

3. **Possible Extensions:**
   - Discussed potential extensions of our proposed algorithm to the offline RL setting and the s-rectangular setting.

4. **Additional Experiments and Explanations:**
   - Provided more explanations about experiment results and conducted an extensive ablation study on DDRQ algorithm's sensitivity to learning rates, showcasing its robustness across different perturbations in both Cartpole and LunarLander environments.

5. **Clarification about Details in the Paper:**
   - Enhanced explanations about definitions, proofs, and the intuition behind our algorithmic design, particularly emphasizing the benefits of the multi-timescale design in terms of training stability and estimation.

6. **Clarification about Assumptions:**
   - Explained the organization of assumptions in our paper, directing reviewers' attention to the mild assumption crucial for our DRRL algorithm's practical value.

We appreciate the support from reviewers ogVe and DkEx after the discussion, and we kindly request other reviewers to review our responses and consider supporting this work if we have adequately addressed your concerns. We are open to further discussions if additional concerns arise.

---

### Meta-Review · Area_Chair_wYVy · 2023-12-06

**Metareview:**

This paper explores model-free distributionally robust RL, where the algorithm learns from a single trajectory in an online manner. The authors propose a distributionally robust Q-learning algorithm, leveraging a multi-timescale approximation scheme derived from existing stochastic approximation results. The paper also presents asymptotic convergence guarantees for the proposed algorithm and includes experiments on simulated problems like Cliffwalking and the American put option environment, as well as classical control tasks such as LunarLander and CartPole.

While the problem addressed is intriguing and novel, the paper lacks an analysis or in-depth discussion of sample complexity, a crucial aspect in the theoretical development of efficient RL algorithms. Reviewers express concerns about the clarity and presentation of the theoretical analysis, noting that there are over ten assumptions in the proof without detailed explanations or explicit calculations to verify that they are satisfied. Additionally, the claim that the R contamination model lies outside the probability simplex is deemed incorrect, casting doubt on other claims.

Moreover, there appears to be a discrepancy in the claim of using deep Q networks, as the experiments seem to involve only two-layer neural networks. To enhance the empirical applicability of the proposed method, the authors are encouraged to conduct additional experiments on more complex environments, such as MuJuCo.

I encourage the authors to diligently address the reviewers' concerns, particularly refining the presentation, ensuring clarity on the proof, and verifying the assumptions used. Furthermore, providing explicit details on the choice of neural networks and conducting experiments on more challenging environments will significantly strengthen the overall contribution of the paper.

**Justification For Why Not Higher Score:**

The paper requires revision to enhance clarity and presentation, particularly regarding the assumptions used in the proof. Additionally, the authors should strive for more accuracy when comparing their method with existing approaches.

**Justification For Why Not Lower Score:**

N/A

---

### Decision · Program_Chairs · 2024-01-16

Reject